# Learning spatial hearing via innate mechanisms

**Yang Chu** [1], **Wayne Luk**[2], **Dan F. M. Goodman**[1]*

**1** Department of Electrical and Electronic Engineering, Imperial College London, London, United Kingdom, **2** Department of Computing, Imperial College London, London, United Kingdom

* d.goodman@imperial.ac.uk

**Data availability statement:** All code and trained network weights are available at

## Abstract

The acoustic cues used by humans and other animals to localise sounds are subtle, and change throughout our lifetime. This means that we need to constantly relearn or recalibrate our sound localisation circuit. This is often thought of as a "supervised" learning process where a "teacher" (for example, a parent, or your visual system) tells you whether or not you guessed the location correctly, and you use this information to update your localiser. However, there is not always an obvious teacher (for example in babies or blind people). Using computational models, we showed that approximate feedback from a simple innate circuit, such as that can distinguish left from right (e.g. the auditory orienting response), is sufficient to learn an accurate full-range sound localiser. Moreover, using this mechanism in addition to supervised learning can more robustly maintain the adaptive neural representation. We find several possible neural mechanisms that could underlie this type of learning, and hypothesise that multiple mechanisms may be present and provide examples in which these mechanisms can interact with each other. We conclude that when studying spatial hearing, we should not assume that the only source of learning is from the visual system or other supervisory signals. Further study of the proposed mechanisms could allow us to design better rehabilitation programmes to accelerate relearning/recalibration of spatial hearing.

## Author summary

The ability to tell which direction a sound is coming from is crucial for the survival of all animals, including humans. We use subtle differences between the sounds heard by our two ears, and these subtle differences are unique to every individual and change during our lifetime. We asked how we can learn these changes, given that usually we don't get feedback from the world telling us which direction a sound came from. This is particularly true of blind people who can't see the object that produced a sound, but can usually tell the direction as well as sighted individuals (and sometimes better!). We found a range of different strategies we can use to reliably learn these subtle cues based on moving our heads around and listening to the same sound multiple times. In particular, an

the following GitHub repository:
https://github.com/YangTrue/
Learning-spatial-hearing-via-innate-mechanisms.

**Funding:** The support of UK EPSRC (grant number EP/X036006/1, EP/V028251/1, EP/S030069/1, EP/L016796/1 and EP/N031768/1) is gratefully acknowledged by YC and WL. The funders had no role in study design, data collection and analysis, decision to publish, or preparation of the manuscript.

**Competing interests:** The authors have declared that no competing interests exist.

innate ability we are all born with can help with this: babies naturally turn their head to the side where a sound comes from, which is something we can do without learning. Our results suggest new experiments that take head movements into account may need to be done to understand how we learn these abilities.

## Introduction

Sensory systems must adapt to changes throughout life to maintain an accurate representation of the environment. The auditory localization system, which enables animals to determine sound source locations, provides an excellent model for studying such sensory plasticity. Neural circuits processing sound localization cues show remarkable adaptability, adjusting to developmental changes like head growth [1] and compensating for hearing impairments [2,3]. However, fundamental questions remain about how the brain accomplishes this complex calibration task.

What could be the calibration signal used by the brain to learn spatial hearing? While the sound localiser can be calibrated (that is, adjusted from an approximately correct starting point, or fine-tuned) via supervised learning using visual feedback as the teaching signal [3,4], considerable evidences indicate that vision-independent mechanisms must also exist [4]. Human can learn to accurately localize sound sources both inside and outside the visual field [5,6], with equal speed and magnitude [7]. Congenitally blind individuals can develop sound localization abilities comparable to, and in some cases superior to, those of sighted individuals [8–10]. Neurophysiological results in animals also demonstrate visual-independent calibration of the auditory space map [11,12]. Several hypotheses [11,12] and computational models [13,14] have been proposed. These hypotheses include the possibility that we make use of cues based on how sounds change under self-motion, audiomotor calibration and feedback from touching objects that produce sounds [9,15,16]. However, the precise mechanisms underlying vision-independent calibration remain largely elusive. Recently, deep learning has emerged as a powerful and flexible tool for modeling sensory systems [17–19], offering new insights into auditory learning [20,21]. However, its effectiveness is limited by prevailing learning paradigms—especially supervised learning—which depend heavily on large volumes of externally provided labels. Moreover, this question of sensory cue calibration in the absence of direct supervision arises in a wide variety of sensory learning contexts and modalities beyond spatial hearing [22–26], calling for a general algorithmic framework to support further inquiry.

We propose "bootstrap learning", a novel type of learning process where innate brain functions, even though basic and minimal, guide the learning of more sophisticated functions without external supervision—analogous to "pulling oneself up by one's bootstraps". Innate neural circuits in the auditory system offer several advantages as the vision-independent teacher for spatial hearing, providing a universally accessible calibration mechanism that is present in every individual throughout life. However, an innate circuit is not sufficient as a localiser given that auditory cues change during a lifetime. To determine the location of a sound, the learned localiser must accurately process complex, direction-dependent acoustic cues. In contrast, innate circuits are often limited to only basic functions, such as crude left-right discrimination, falling far short of offering precise localization supervision. Could bootstrap learning truly be feasible for spatial hearing?

We utilize simulations to examine the bootstrap learning principles, integrating three core components: a small "Teacher" neural circuit with basic innate functionality, a plastic "Student" neural network with sufficient capacity to learn the complex sound localization function, and an interactive acoustic "Environment" (Fig 1). Both the Teacher and Student are internal components of an Agent's brain. The "Agent"—defined here as any human, animal, or model capable of acting within the environment—receives auditory inputs and moves in the simulation. The Student is a deep neural network that models the auditory space map and related neural systems in the brain, which we refer to as a "sound localiser" in this paper. The Teacher is a much simpler, hardwired neural circuit that provides internal calibration signals for the Student. Bootstrapping in this context resembles a blindfolded single-player game: an Agent must learn the accurate localiser through exploration, relying solely on its innate Teacher circuit as self-guidance, without access to visual feedback or external labels. This is one way of implementing the hypothesis of calibration by audiomotor feedback [9,14]. To assess the plausibility of different candidate Teacher circuits, we evaluate how effectively each guides the Student's learning. Additionally, we analyze the computational principles of bootstrapping using systematic simulations, exploring how simple innate mechanisms can facilitate the development of more complex neural functions.

## Results

### Innate LSO circuits can bootstrap a 360 degree localiser without external supervision labels

Newborns turn their head toward sound sources. This innate behavior is named the Auditory Orienting Reflex (AOR) [27]. Although newborns can not accurately localize sound sources, their AOR is relatively accurate for left-right discrimination [28]. We investigated whether or not a simple neural circuit model of the AOR would allow "bootstrap learning" of an accurate 360° sound localiser in the azimuth plane—without external error feedback or supervision labels.

There are two neural modules in our model. The first—the Teacher—models the basic AOR as a small neural circuit that discriminates whether a sound source is on the left or right side. The second part—the Student—models a complete sound localiser as a deep neural network (DNN) (Fig 1), which consists of 5 fully-connected hidden layers, with each layer containing 128 perceptrons with ReLU activation functions.

Previous approaches to training deep neural networks (DNNs) for predicting the direction of sound arrival typically rely on supervised learning, which involves minimizing a regression loss $J$ with a labeled dataset $(s, y^*)$ [20,21]. In this paradigm, each data point comprises a sound $s$ originating from a direction $y^*$, where $y^*$ is a real-valued variable representing the true direction of arrival. The label set must be sufficiently large to encompass the full range of possible sound-direction combinations. The loss function $J$, such as the L1 distance $|y^* - \hat{y}|$, measures the discrepancy between the predicted direction $\hat{y}$ and the true label $y^*$, and its gradient is used to update the DNN parameters. Biological supervised learning for spatial hearing in the brain [3,4] may not rely on this exact gradient update at the neuronal level, but they do require visual feedback—originating outside the auditory system—to provide a precise supervisory signal (the label $y^*$) that must be matched by the sound localiser.

Can the Student DNN learn a complete 360° sound localiser using only the basic left/right classifier as the Teacher, without access to any external label $y^*$? The supervised learning approach is not applicable in the absence of $y^*$. To explore the bootstrap learning principle, we first propose an interactive learning procedure assuming an abstract Teacher circuit, then evaluate this procedure with various Teacher circuit implementations through simulation.

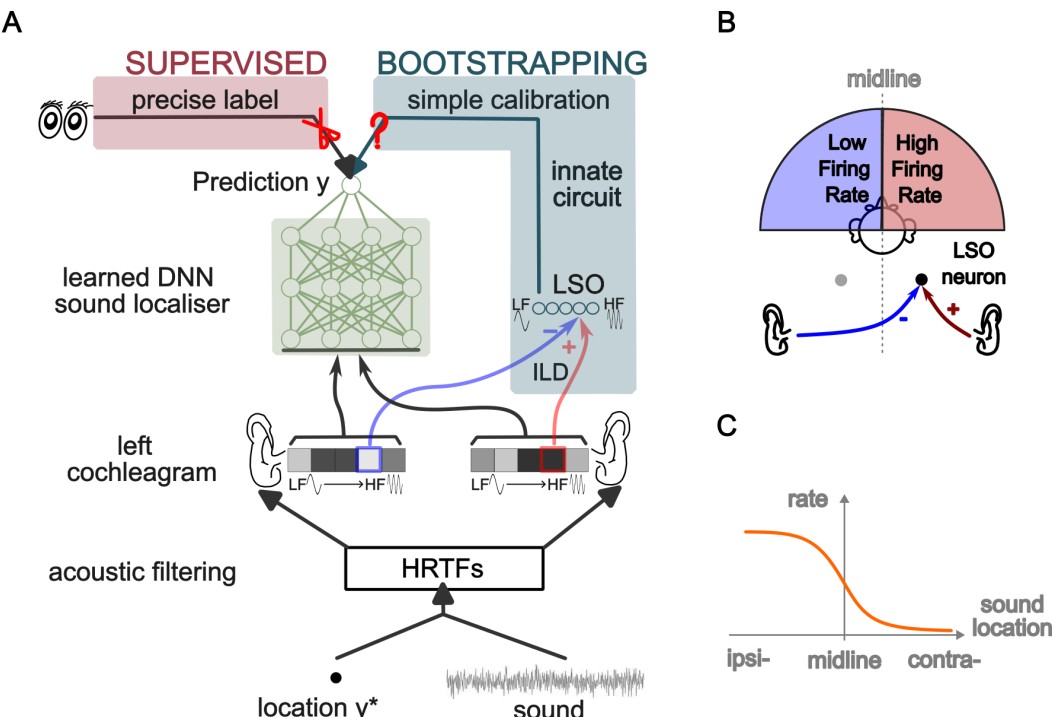

**Fig 1. Overview.** The aim of spatial hearing is to estimate the position of a sound source using acoustic cues. The sound localiser needs to be learned and continuously re-calibrated over the individual's lifetime. (A) Different learning paradigms and the overall model of the Agent for spatial hearing. The acoustic environment is simulated using pre-recorded head-related transfer functions (HRTFs), converting a sound stimuli into a cochleagram (cochlear responses in each frequency band) and fed to a small innate "Teacher" circuit and a bigger plastic "Student". The Student is implemented as a deep neural network and trained with coarse-grained feedback from the noisy Teacher, a process we refer to as "bootstrapping". (B) One of the Teacher circuits we use is an abstract model of the lateral superior olive (LSO) receiving excitatory input from one side and inhibitory input from the other. (C) Typical response tuning curve of an LSO unit to different sound locations, showing its basic spatial hearing function as a left/right classifier, but not accurate enough as the sound localiser which should predict the exact angular value $y$. Abbreviations: LF–Low Frequency. HF–High Frequency. ILD–Interaural Level Difference. DNN–Deep Neural Network.

In this procedure, the Teacher's binary left/right feedback is used to approximate the direction of adjustment of the current prediction (which turns out to be mathematically equivalent to approximating the gradient of the L1 loss for regression, from the statistical learning perspective).

The interactive procedure consists of the following steps:

1. At time step $t$, a sound is presented from an unknown location (angle) $y^*(t)$ on the azimuth plane. The agent uses the Student network to make an initial prediction $y(t)$.
2. The agent rotates toward its predicted direction $y(t)$. Current angle distance between the sound source and the agent's orientation changes to $y^*(t+1) = y^*(t) - y(t)$
3. The agent listens again, using the Teacher to decide whether the sound source is to the left or right at the new position:

$$\text{sign}(y(t+1)) \approx \text{sign}(y^*(t+1)) \tag{1}$$

where the Teacher output is $\text{sign}(y(t+1))$, a binary variable.

4. The Teacher output is used to approximate a surrogate gradient of the error $J$ with respect to the Student parameters $\mathbf{w}$.
5. The approximated gradient is then plugged into stochastic gradient algorithms to adjust the Student parameters.
6. This is repeated until the Student converges.

In the above equations, the angle $y \in [-180°, 180°]$ and sign function are defined as:

$$\text{sign}(y) = \begin{cases} -1, & y < 0, \text{ sound on the left, or anticlockwise difference} \\ 0, & y = 0, \text{ sound at the front, aligned with the midline} \\ +1, & y > 0, \text{ sound on the right, or clockwise difference} \end{cases}$$

The approximation in Eq (1) is due to the biological inaccuracy and stochasticity of the Teacher circuit, which does not always provide correct feedback.

The surrogate gradient is:

$$\frac{\partial J}{\partial \mathbf{w}} = \sum \text{sign}(y^*(t) - y(t)) \approx \sum \text{sign}(y(t+1)) \tag{2}$$

Although the proposed "surrogate" gradient algorithm looks unfamiliar, the gradient update is actually equivalent to using an L1 loss function, or mean absolute error (MAE), but eliminates the dependency on the exact angle values needed in supervised learning. The L1 loss, a common regression objective for supervised learning models, is known for its robustness to outliers compared to L2 loss (mean squared error), though it typically converges more slowly as the error approaches zero. Our algorithm requires no more assumptions than supervised learning with gradient descent as a model of biological learning [17].

The above interactive procedure uses an abstract Teacher, but does not specify the exact implementation of the Teacher neural circuit. We now describe two different Teacher circuit implementations.

Lateral Superior Olive (LSO) neurons, which are sensitive to interaural level differences (ILD) and serve as basic left/right sound discriminators in the brainstem, have been suggested to be involved in the Auditory Orienting Reflex (AOR) [29,30]. The LSO receives excitatory input from the ipsilateral ear and inhibitory input from the contralateral ear, allowing it to compare sound intensity levels between the two sides. When the sound originates from the ipsilateral side, the LSO responds with high spiking rate, whereas it remains silent for sounds originating from the contralateral side (Fig 1). The LSO neuron has a stochastic sigmoidal response to ILD, approximating but not exactly matching the ideal Teacher—a step-function (the sign function in Eq 2) which is the perfect left-right discriminator. The LSO exhibits several limitations in spatial encoding. First, it cannot resolve front-back ambiguity due to simple circuit structure, which prevents the integration of cues beyond ILD, such as broadband spectral information. Second, its sigmoidal response function saturates at more lateral positions as the sound source moves away from the midline, limiting the ability to precisely decode angular information from its output. Finally, even near the midline, the stochastic nature of LSO responses also lead to left-right classification errors (Fig 2).

We first implement a single LSO neuron Teacher model based on an empirical dataset of mammal LSO neurons [31], characterized by their mean and variance of the response firing rates to different ILDs. Note that we chose to use only ILDs rather than interaural time differences (ITDs) to keep the model's neural network architecture simple and to avoid the use of spiking neural networks which are more challenging to train, although recent research

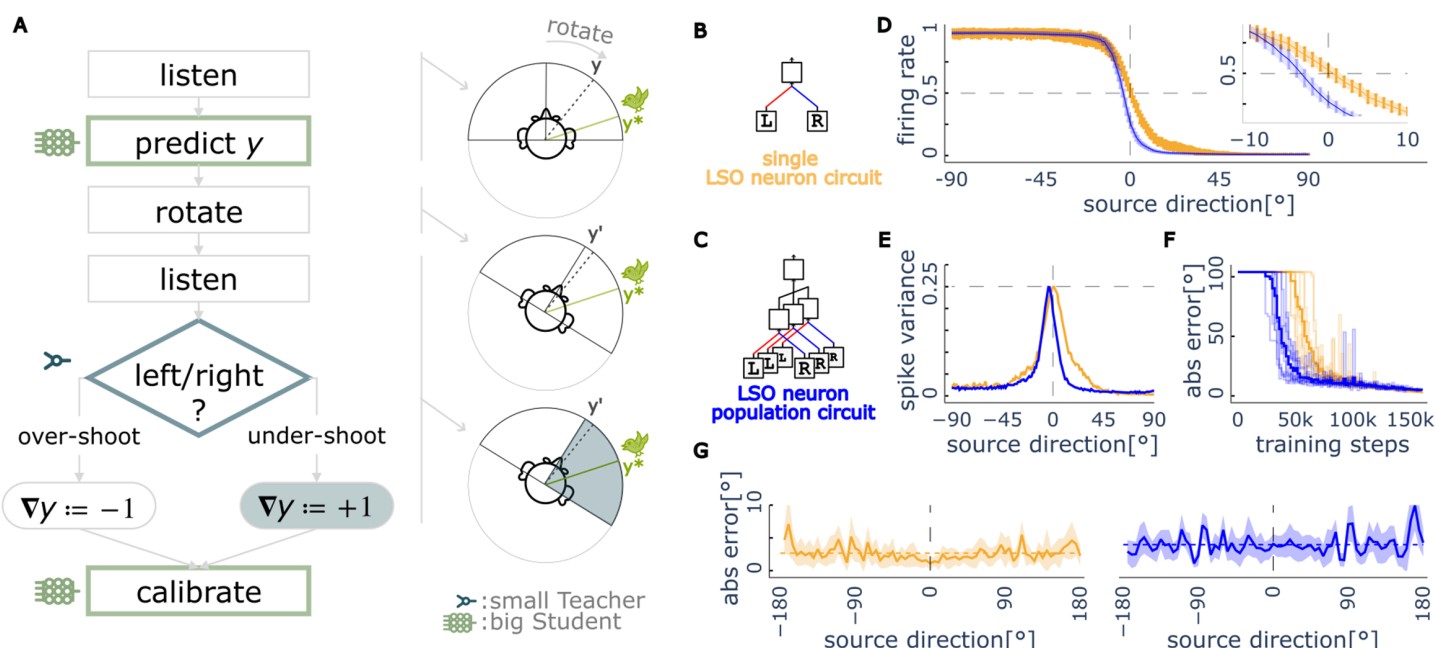

**Fig 2. Bootstrapping spatial hearing from an innate circuit.** (A) Interactive learning procedure with the left/right Teacher circuit. The agent makes an initial prediction of the sound location with its learned Student network, turns its head towards that sound, and uses the coarse-grained feedback (left/right) from the Teacher circuit to update the learned localiser based on whether it undershot or overshot the target. (B) A Teacher circuit implementation using a single lateral superior olive (LSO) neuron as the left/right discriminator (C) Another Teacher circuit using a population ensemble of LSO neurons. (D) Mean normalized firing rates of two Teachers—single LSO neuron (yellow) and LSO neural population (blue)—as functions of sound source angle, with variance indicated by vertical bars. Inset shows responses near midline (0°), where LSO neural population Teacher exhibits a slight leftward bias in the 0.5 firing rate crossing point, while the single LSO neuron Teacher shows a minimal rightward bias. (E) Response variance across sound positions. Both Teachers approach theoretical maximum Bernoulli variance (0.25) near their respective midline positions and minimal variance at lateral positions, indicating increased uncertainty for left-right discrimination at positions approaching the midline. LSO neural population Teacher shows narrower variance peaks and lower variance magnitude compared to single LSO neuron, with peak variance position reflecting the same directional biases observed in the mean responses. (F) Learning trajectories for Student networks trained with each Teacher type (mean shown as solid line, individual repeated experiments as shaded lines). All achieve mean absolute errors (MAE) below 5° after training, with LSO neural population Teacher enabling faster initial convergence but slightly higher final error. (G) Spatial distribution of localization errors after training (solid lines show mean across runs, shaded regions show ±1 SD). Both Teacher types enable precise mapping across all positions, with lowest error near the midline and increased errors at ±90°. Students that learns from the LSO neural population Teacher show marginally higher average error (dashed horizontal lines), consistent with the Teacher's inherent directional bias.

attempts to address this issue [32,33]. We choose a single LSO neuron with a 2000 Hz characteristic frequency as our Teacher circuit. The Teacher circuit generates stochastic binary feedback (a single spike or not) based on the tuning curve. We evaluate this single LSO neuron circuit as the abstract left-right discriminator Teacher in the bootstrapping procedure described above.

The experiment comprised 10 repetitions. During training, sounds were sampled randomly from the full 360° horizontal plane, with random level fluctuations (±20 dB) added to simulate natural amplitude variations. Trained networks were tested on 72 equally spaced locations (5° intervals) from 0° to 360° (see more details in Methods).

Results after training show an average mean absolute error (MAE) of 2.7° between predicted and actual angles, with a mean prediction variance of 1.7° (Fig 2). The magnitude of errors at midline matches human behavioral studies well, although human performance at the periphery is much worse than the predictions by our model [6,34].

We next examined another circuit model of the Teacher, by averaging the output of a population of LSO neurons with characteristic frequencies spanning from 20 Hz to 2200 Hz (see

more details in Methods). This ensemble circuit also outputs a stochastic binary feedback (firing a spike or not) via a sigmoid readout neuron, but offers two key advantages comparing to the single LSO neuron circuit: a broader frequency response range and reduced signal variance through population averaging, resulting in more reliable learning signals.

Using identical configurations as the single LSO neuron Teacher, Students learned from this LSO neuron population Teacher achieved a mean absolute error (MAE) of 4.0° with a prediction variance of 2.0°. Compared to learning with a single LSO neuron, learning with the ensemble is faster (Fig 2F) due to the reduced variance of the teaching signal (Fig 2E) (the ensemble reaches 10.0° error at 94K training episodes, while the learning with the single LSO neuron Teacher requires 10K episodes longer). However, learning with the ensemble Teacher has a slightly larger final error (Fig 2G), because this ensemble Teacher has a larger bias at the midline (Fig 2D). We will return to these points in the next section.

In summary, these results demonstrate two key points. First, a single LSO neuron circuit can learn a complete 360° sound localiser via bootstrapping, without requiring externally provided labels, despite the LSO's inability to process spectral cues or to distinguish front from back. This illustrates how a simple Teacher circuit can guide the Student network to learn a more sophisticated mapping than the Teacher itself. Second, because bootstrapping only requires the Teacher to provide a basic functional signal without high precision demands, a wide range of neural implementations can serve this role equivalently. This flexibility results in a large and interchangeable set of candidate circuits, making the bootstrapping principle both broadly applicable and robust.

## Learning accuracy depends on Teacher bias

We quantitatively assess the relationship between the performance of different Teacher circuits and the learning outcomes of their corresponding Student networks. The Student model learns to predict sound source direction, which we measure by its accuracy (mean absolute error). The LSO Teacher gives a binary discrimination (left/right), which we measure by its bias and variance. Bias reflects the deviation of the decision boundary from the true midline. A positive bias indicates a rightward shift of the boundary, leading to sounds from the right being misclassified as originating from the left(counter-clockwise error). Conversely, a negative bias results in a leftward shift, causing sounds from the left to be misclassified as coming from the right(clockwise error). Variance captures the Teacher's prediction uncertainty across sound locations (Fig 2E). Although all LSO neurons share similar innate circuit connectivity, natural variations in their responses exist. These circuits can become biased due to sensory changes, such as hearing impairment, or initial configuration differences arising from genetic variability.

Using recorded data from 32 LSO neurons [31], we found that the bias of the LSO Teacher (acting as a left-right discriminator) is positively correlated with the accuracy of the Student (learned DNN): Students trained with less biased Teachers achieved higher accuracy (Fig 3). This finding aligns with the mathematical equivalence of the surrogate gradient estimation Eq (2) to a shifted regression target function. Intuitively, if the Teacher exhibits a consistent error of 5°, the Student cannot learn with error lower than 5° and will eventually converge to a localiser with the same systematic offset. In addition, larger Teacher variance consistently led to slower convergence of the Student network, due to the stochastic gradient-based optimization algorithm used for training. While the learning process remained stable and eventually converged across a wide range of Teacher variance values, it failed under extreme noise in 4 out of 96 simulations with current experiment setting. The exact convergence rate depends on multiple factors—including the biological plausibility of gradient-based learning

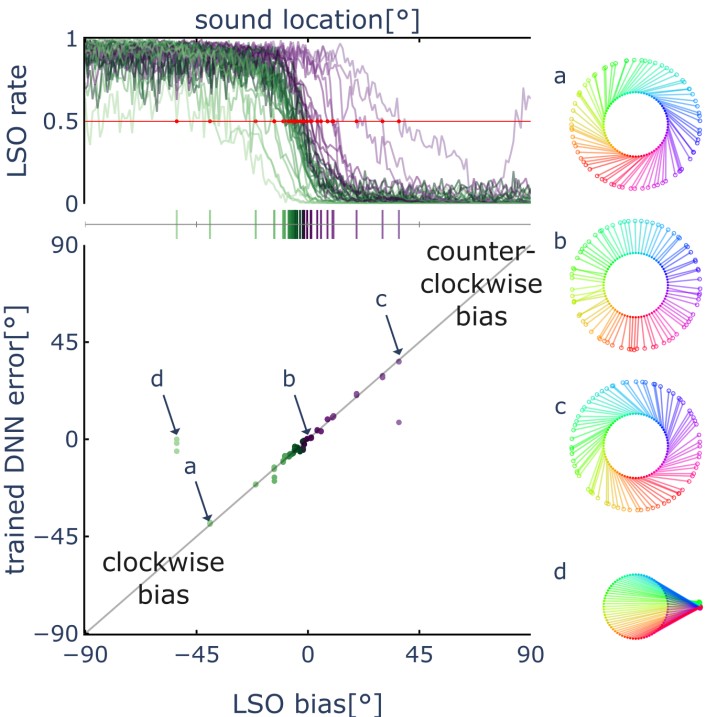

**Fig 3. Student error follows Teacher bias.** Top plot: tuning curves of 32 lateral superior olive (LSO) neurons for different sound source locations, color-coded by their estimated bias (code shown in the vertical line segments below). While most neurons exhibit small biases near the true midline, several show larger deviations, demonstrating natural variability for different Teachers with the same innate circuit connectivity. Bottom plot: the estimated Teacher bias matches the trained DNN Student test error (measured here as signed error, positive for clockwise direction difference, negative for counter-clockwise). Four out of 96 cases failed to converge within the allowed training period due to extremely noisy Teacher signals that slows down Student learning. Right hand plots: learned maps for four representative cases (a-d). Each point on the inner circle represents the true angle on the horizontal plane, while corresponding points on the outer circle show the angle predicted by the Student model. Connecting lines are color-coded by true angle. Case b (LSO neuron no.24 in [31]) shows the ideal radius-aligned lines, indicating accurate learning. Cases a and c (neurons no.1 and no.32) demonstrate systematic clockwise or counter-clockwise bias respectively, illustrating shifted Student maps bootstrapped from biased Teachers—similar to shifted auditory maps learned with visual prism adaptation. Case d is one of the 4 unconverged cases due to high variance in LSO Teacher signal.

mechanisms—though modeling biological learning speed is beyond the scope of this study. Importantly, the final performance of the Students do not correlate with the Teachers' variance after sufficient training. Overall, the results show that a basic Teacher providing moderately accurate, albeit noisy, guidance is sufficient to support robust learning of the Student network.

## Innate LSO bootstrapping facilitates adaptation after changes of acoustic cues

The brain faces the continual challenge of adapting to shifts in sensory input. However, it is impractical to generate new supervised label datasets each time such sensory changes occur, especially given their frequency and unpredictability throughout development and beyond. Deep neural networks (DNNs), used as models of the brain and trained under the supervised

learning paradigm, encounter the same challenge. In fact, DNNs are often even more sensitive to cue disruptions [35]. It is known that small perturbations in the input—sometimes imperceptible to humans—can lead to catastrophic failures in DNN predictions [36].

We investigate how an agent can effectively maintain spatial hearing when alterations to acoustic cues occur. We begin by introducing a range of sensory disruptions and independently evaluating the sensitivity of both the innate LSO Teacher circuit and the learned DNN Student. This comparison highlights their distinct responses to changes in acoustic cues. We then reapply the bootstrapping procedure to evaluate its capacity for self-sufficient adaptation. Note that that we do not address the much slower changes to acoustic cues during development, as head size changes.

In our model, the peripheral auditory system feeds relative neural response magnitude vectors $(L, R) = (L^\eta, R^\eta)$ as the input for both Teacher and Student (see Methods). We model sensory changes in hearing via a combination of shifting (additive) and scaling (multiplicative) of these magnitudes. We tested three scenarios: symmetrical shifting ($L' = L - 20, R' = R - 20$), where $(L, R)$ are the original inputs before sensory disruption, symmetrical scaling ($L' = 0.5L, R' = 0.5R$), and asymmetrical scaling ($L' = 0.25L, R' = R$). We could also have considered asymmetrical shifting (e.g. $L' = L - 20, R' = R$), however since this just introduces a bias in the Teacher, this would have a similar effect as asymmetrical scaling (Fig 4 top right

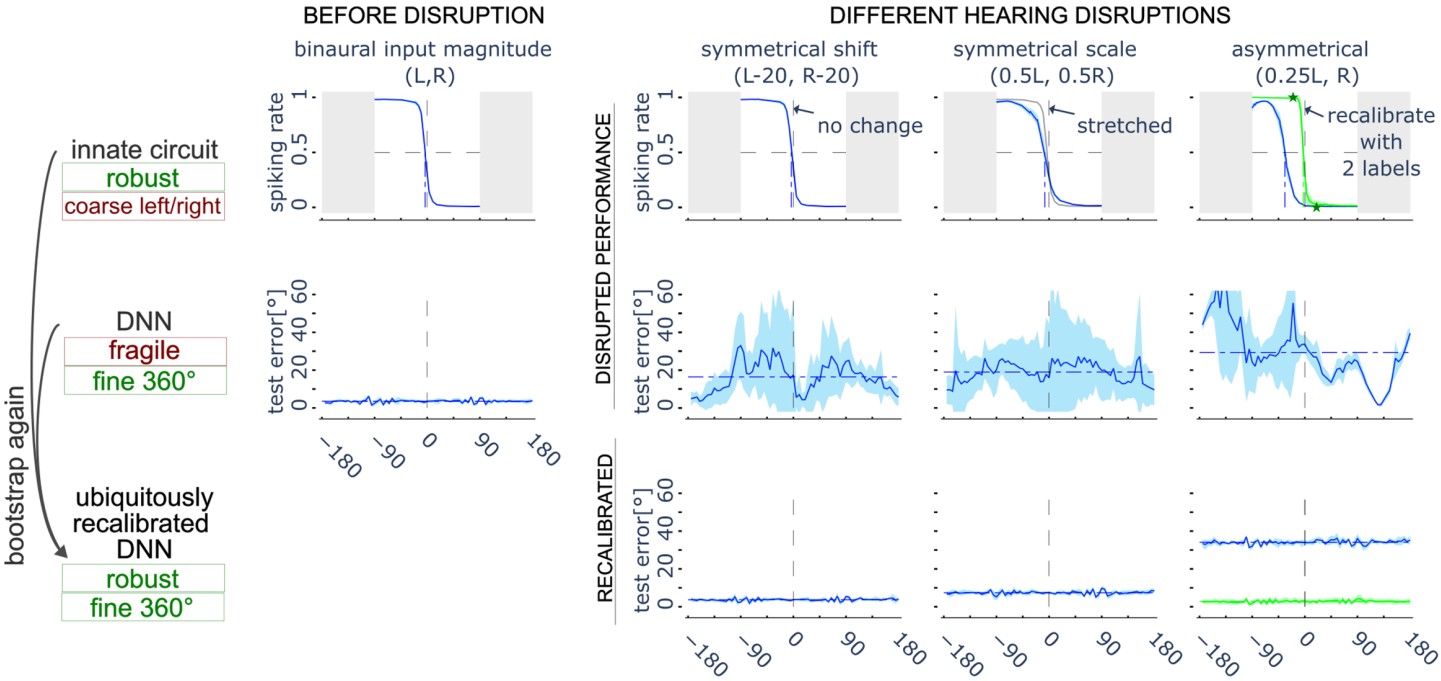

**Fig 4. Effects of cue disruptions and re-calibration mechanisms.** Response of the Teacher circuit (top row) and DNN Student (middle row without bootstrapping; bottom row with bootstrapping). The left column shows the original ILD response curve of the Teacher (top) and the good performance of the Student (bottom) before any acoustic cues are disrupted. The three right columns show the effects after three different types of disruptions to the acoustic cues. A symmetrical shift (left column, symmetrical bilateral hearing loss) leaves the ILD sensitivity curves (top row) of the Teacher unchanged. A symmetrical scaling (middle column, symmetrical bilateral auditory compression disruption) stretches the response curve along the ILD axis but doesn't change the bias (preference for left/right). An asymmetrical scaling (right column, asymmetrical unilateral hearing loss), changes the bias of the LSO curve, although this can be restored with two labeled data points (green curve). In contrast, the DNN Student is much more sensitive to any disruptions in acoustic cues. The Student prediction is initially disrupted with high errors (middle row), but after recalibration (relearning using the Teacher) good performance is restored for the symmetrical disruptions (which do not change the bias). In the case of the asymmetrical disruption (right column), performance is restored after recalibration of the Teacher (green curve, bottom row).

panel). These conditions simulate possible components of hearing loss or disrupted auditory signal compression across a broad frequency range. In reality, hearing changes often involve complex, frequency-dependent combinations of both shifting and scaling transformations, as well as nonlinear effects, which typically require detailed empirical measurements to characterize precisely. Our computational model enables the isolation and analysis of some of the abstract components of these transformations individually, allowing us to cleanly examine their effects.

We start with the effect of these disruptions on the Teacher (Fig 4), the ensemble LSO left-right discriminator. For the symmetrical shifting, the ILD does not change, so the Teacher is not affected at all. For the symmetrical scaling, the ILD is uniformly scaled, resulting in a corresponding scaling of the Teacher's response curve along the ILD axis. This scaling stretches the response symmetrically in both directions around the midline. The center of this transformation is the midline itself, and the new bias is given by $\beta' = C \times \beta$, where $C$ is the scaling factor. Consequently, an unbiased Teacher (i.e., $\beta = 0$) remains unbiased after scaling, as $\beta' = C \times 0 = 0$. Similarly, a Teacher with a small initial bias ($\beta \approx 0$) will retain near-accurate performance after scaling, since $\beta' \approx 0$ as well. For the asymmetrical disruption, the ILD is shifted and then the Teacher's response curve is also shifted. The Teacher's bias increases if the shift is substantial, meaning the Teacher will now be inaccurate. However, if we allow plasticity in the readout neuron of the Teacher (Methods), it can be re-calibrated with as few as two label-stimulus pairs. Note that the two labels are simply binary labels of left or right, which does not require exact angle values. The variance in the Teacher's output is not significantly affected by the input disruption, as it is primarily determined by the inherent stochasticity of the biological circuit model. In summary, the Teacher is robust to these sensory disruptions after at most a minimal amount of relearning.

In contrast to the robust Teacher, the Student is highly sensitive to changes in input cues: even these relatively small sensory alterations in our simulation lead to large shifts in the Student's predictions, with both prediction error and variance increasing significantly (Fig 4). This not only highlights the DNN's sensitivity to input distribution shifts, but also reflects the fundamental challenge faced by the brain—balancing the need to remain sensitive to subtle acoustic cues for accurate spatial inference, while maintaining robustness against common sensory perturbations.

Next, we combine both the Teacher and Student within the bootstrapping framework and carry out a self-sufficient recalibration process by repeating the same learning procedure described in the previous section (Innate LSO circuits can bootstrap a 360 degree localiser without external supervision labels). In cases of symmetrical shifting or symmetrical scaling, the Teacher remains robust and nearly accurate, allowing the Student to be accurately re-learned without the need for any external labels. However, under asymmetrical scaling, the non-plastic Teacher exhibits a large bias after disruption. As a result, although recalibration reduces the variance in the Student's output predictions, the learned Student inherits the Teacher's bias—consistent with previous analysis (in Learning accuracy depends on Teacher bias). This issue is relatively straightforward to address with minimal intervention. A plastic Teacher can serve as an intermediary, enabling effective recalibration of the Student using only minimal external supervision—in this case, just two binary-labeled data points. This approach addresses two critical limitations of direct supervised learning. First, binary left/right labels cannot be used for training a fine-resolution Student localiser in the regression task, which requires real-valued azimuth angles as ground truth. Second, a mere two labeled data points are inadequate for learning the full spatial range via supervised methods. The Teacher, acting as a mediator, thus fulfills a dual role: it reduces both the quality and quantity

of external supervision required for successful adaptation. Both the accuracy and variance of the Student are restored after the recalibration (Fig 4).

Neither the simple Teacher nor the more complex Student alone meets the dual requirement of robustness to input changes and the ability to extract fine-grained features necessary for accurate spatial localization. However, bootstrapping offers a powerful integration of both: it preserves the Student functional accuracy through self-sufficient adaptability, without the need for extensive external labeling. As a result, recalibration can be achieved via bootstrapping using an innate circuit, available at any time, enabling adaptation even in the absence—or scarcity—of external supervision in real-world settings.

## An innate circuit can provide the intrinsic reinforcement reward

Owl's auditory space map exhibits enhanced plasticity when hunting live prey compared to when they are fed dead mice [37]. This observation aligns with reinforcement learning models, where positive or negative rewards for the animal's actions are provided by the environment. Nonetheless, the application of standard reinforcement learning models is constrained by practical challenges, as external rewards—such as successful prey capture—are typically sparse in real-world environments. Furthermore, ferrets can calibrate their sound localization even after the removal of external rewards, suggesting learning can become self-sustained [12]. Could there exist intrinsic "reward" signals—internally generated and continuously available—that support more flexible and autonomous learning, independent of sparse external rewards?

Neurophysiological evidence suggests a potential source of such intrinsic reward. Neurons preferentially responding to midline sound sources (zero ILD) have been identified in the inferior colliculus (IC) and other regions throughout the ascending auditory pathway [38–41]. These midline detectors exhibit relatively broad spatial tuning, with elevated firing rates not only for sounds originating precisely at the midline (zero ILD) but also for nearby locations. In contrast, external rewards offer higher spatial precision, they are not always available in natural contexts.

We propose and evaluate an alternative auditory orientation response (AOR) model for spatial hearing learning, based on a reinforcement learning procedure and a circuit model of midline-detecting neurons in the inferior colliculus as the internal Teacher. The Teacher fires a spike—providing a positive reward signal—when the agent faces the sound source direction.

In this new interactive procedure (Fig 5 and Algorithm 2) with reinforcement learning, for each episode, when the agent detects a sound, it makes an initial prediction $y_1$ using its current Student mapping and rotates to face that predicted direction $a_t = y_1$. After rotation, if the agent's midline is aligned with the sound source, the internal Teacher provides a positive reward signal, otherwise a negative or zero reward. The Student can then update its parameters based on the reward.

This procedure does not require any externally provided rewards. Similarly to the left/right discriminator procedure, the agent also tries to rotate toward the sound source. But, in this case, it combines the internal Teacher circuit with a reinforcement algorithm - rather than a surrogate gradient algorithm - to adjust its initial spatial prediction. Among the various reinforcement learning algorithms available for reward-based parameter updating, we adopt the policy gradient algorithm as a simple and effective choice (Algorithm 2).

We propose a simple circuit model based on known anatomical connectivity for the inferior colliculus (IC) (Fig 5B). The model consists of two lateral superior olive (LSO) population circuits - one for each side of the brain - whose outputs are multiplicatively combined. As

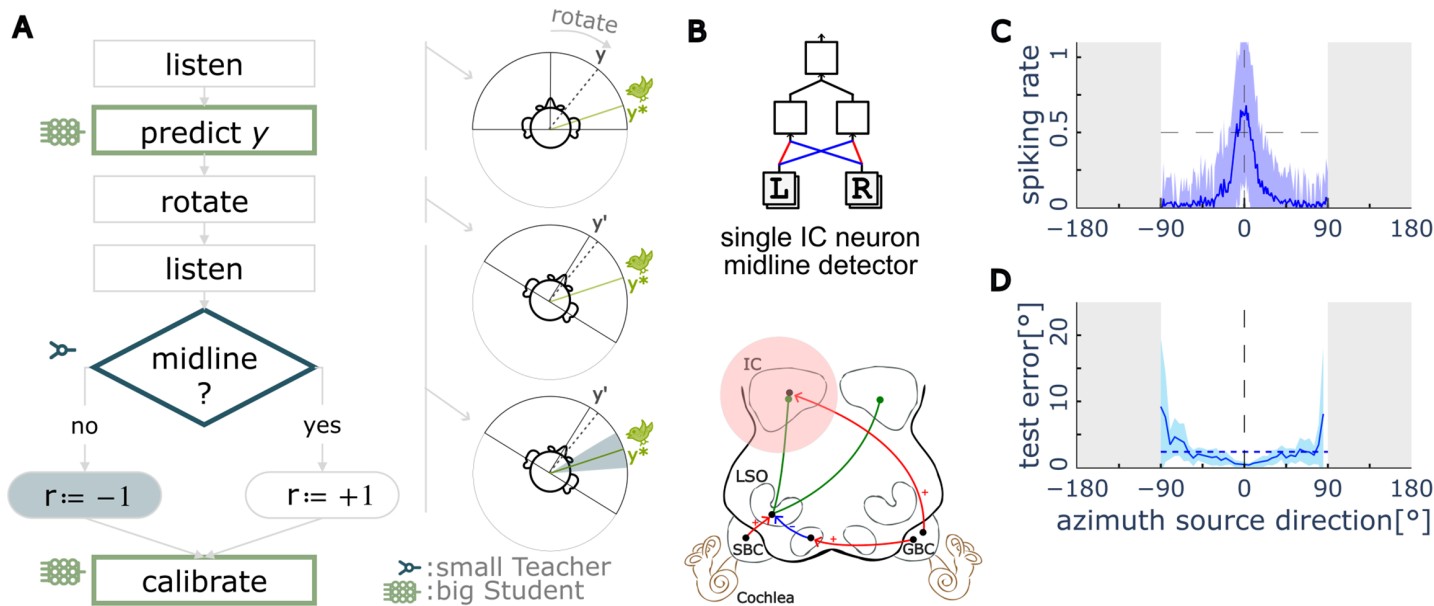

**Fig 5. Innate circuit detecting midline alignment as an intrinsic reward.** (A) An interactive procedure of using the innate Teacher circuit to detect the midline alignment, used as the intrinsic reward signal for reinforcement learning without any external labeling. (B) The Teacher circuit implementation, where the left LSO output and the right LSO output are combined. Circuits with similar connectivity and tuning curves have been found in the inferior colliculus (IC). (C) Sampled tuning curve of the Teacher circuit, showing the basic function as a midline detector - fires when the agent faces the sound. In the reinforcement learning procedure, output spiking means positive reward, no spiking means no reward, offering an alternative model of auditory orienting response(AOR). (D) Test errors of the Student after training, in the frontal semicircle

shown in its response curve (Fig 5C), the circuit exhibits peak activity when the sound source is near the midline, with reduced responses for more lateral sound locations. This behavior is consistent with biophysically observed spatial tuning curves along the auditory pathway [38–41]. This model thus offers a simplified circuit-level account of how binaural information may be integrated in the IC. Binary samples from this response function are then used as the reward signal for the reinforcement learning algorithm.

After training, the Student's predictions achieve a mean absolute error of just 2.4° within the frontal semicircle. Higher errors are observed at the lateral extremes, which may be attributed to two contributing factors. The deep neural network is initialized to predict angles near the midline, making learning in that region inherently easier. In addition, the acoustic characteristics of sound localization at lateral positions introduce greater ambiguity, further complicating accurate learning in those regions.

Like the left-right discriminator, a midline detector Teacher circuit can be characterized by its bias and variance at detecting midline alignment. Note that in this case, the Teacher variance may actually benefit reinforcement learning by enabling better exploration. The intuition is that a very precise detector which only rewards exact midline alignment would make initial learning difficult, since positive rewards would be rare. The Teacher circuit's inherent variance thus provides intermediate rewards that help guide exploration of the spatial mapping.

## A spherical localiser can be learned via bootstrapping in challenging conditions

So far, we have explored various learning algorithms and implementations of the Teacher circuit. We now investigate whether this flexibility of bootstrapping can facilitate learning in

even more challenging contexts: learning to map 3D auditory space using only a minimal monaural Teacher circuit and a restricted range of head rotation. We begin by introducing three variations of the learning procedures and then evaluate their combined effectiveness.

First, we constrain the agent's rotational range to a small window (here, limiting rotation to a maximum of 30° in either direction). We have assumed that the agent can freely rotate toward the sound source; however, this assumption may not hold in all situations. It is often difficult for the agent to rotate fully, particularly to the back, or there may be insufficient time to rotate from one direction to another. To account for these limitations, a dynamic bootstrapping process can be constructed, in which the Teacher is not always fixed; instead, an array of Teachers can be progressively constructed (Fig 6). Initially, the innate circuit is used as the Teacher, which helps the agent learn the localiser within its limited motion range (i.e. only learning to localise sounds from −30 to +30 degrees). This can be done by using any one of the previous models. In the next phase, the localiser learned in the first phase takes the role as the new Teacher: when hearing a sound, the agent can make a first prediction $y_1$, then if $y_1$ is out of the rotation range, the agent can first rotate toward the sound until its rotation

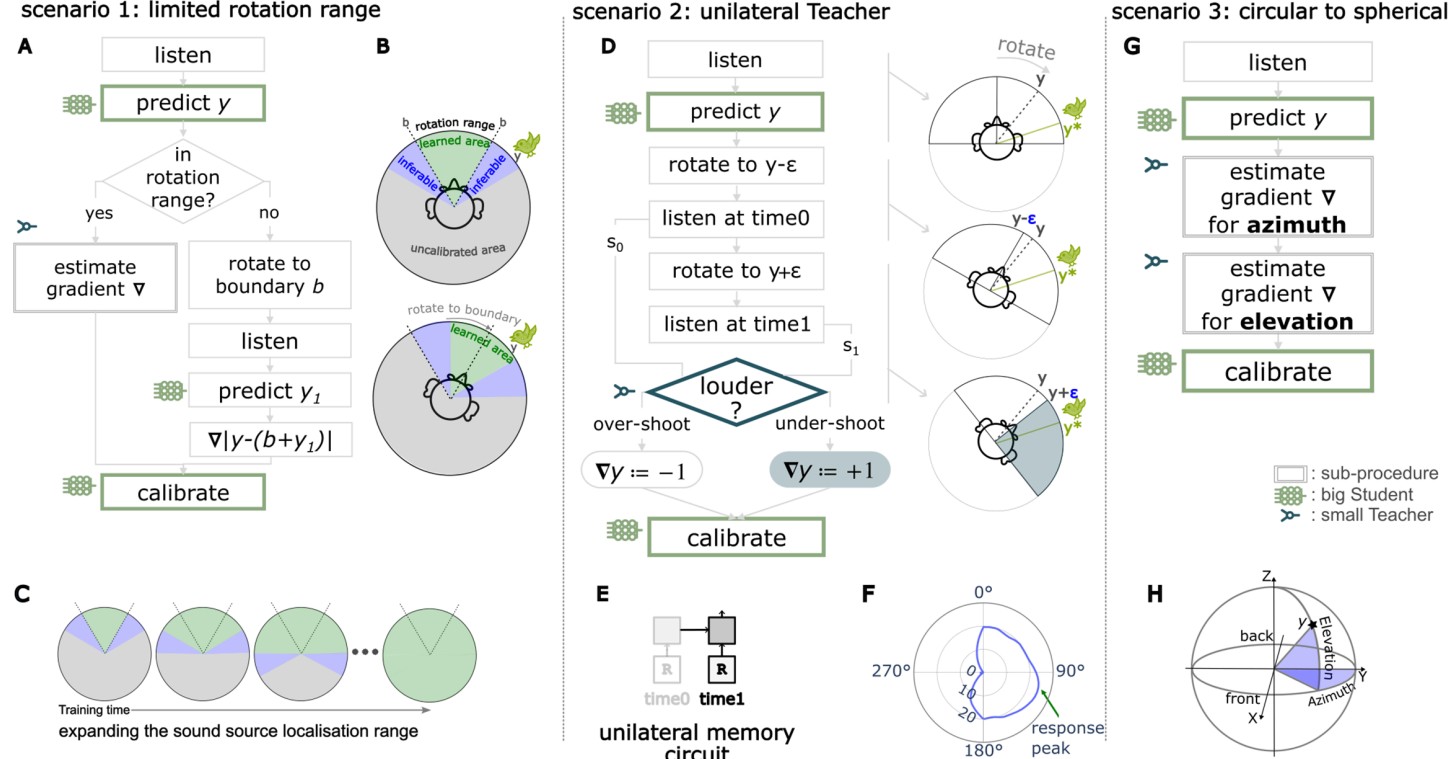

**Fig 6. Bootstrapping procedures for different learning contexts.** Scenario 1: bootstrapping with a limited head rotation range. (A) The agent estimates sound locations within the rotation range using an innate Teacher, and estimate sound locations beyond the rotation range using a learned localiser as Teacher. Gradient estimation with innate Teacher can be done by any method, e.g. surrogate gradient or reinforcement learning. (B) Expanding the already learned localisation range (green area) into the "inferrable area" (blue) by combining head rotation and bootstrapping on existing localisation range, constructing an array of Teachers and Students during the process, by replacing the old Teacher with the learned localiser after each expansion. (C) The process by which the learned part of the localisation range expands to fill the whole space during training. Scenario 2: using a unilateral rather than bilateral Teacher circuit. (D) The bootstrapping process, showing how a gradient can be estimated with a unilateral circuit by pointing the right ear toward the sound source to find the loudest response. (E) The unilateral circuit with an abstract memory unit. (F) 2D response map of the right ear, showing the peak response when the right ear is pointed toward the sound. Scenario 3: extending from a circular localiser to a spherical localiser. (G) The process by which the circular procedures can be extended to spherical procedures by composing them for azimuth and then elevation. (H) Illustration of azimuth and elevation.

limit $\ell$, and then listen again, using the learned localiser to make a second prediction $y_2$. The final prediction from the Teacher will be $y_2 + \ell$, which can be used to adjust the first prediction $y_1$, beyond the the rotation range. This allows the localiser to expand to range between $[-60°, +60°]$. The Teacher is then replaced with this new localiser. Such expansion process can be repeated, with each new Teacher covering a wider range than the previous one, growing the learned localiser by 30° each time until the full range is learned.

We next test a monaural Teacher circuit without relying on the bilateral acoustic cues but instead using simplified monaural spectral cues. We propose a peak-seeking procedure in this case, instead of the midline-seeking procedure used in the previous models. Intuitively, the agent can "point its ear toward the sound source by finding the loudest direction". When the agent hears a sound, it first predicts the angle as $y$. To verify the first prediction, the agent needs to compare the adjacent locations, by moving its head to the left $y - \epsilon$, listening again with response $s_{-\epsilon}$, and then to the right $y + \epsilon$ with response $s_{+\epsilon}$. By comparing the responses at these three locations, the agent can determine if the first prediction needs to be adjusted to the left or right, by following the louder sound. The peak-seeking procedure requires comparisons of the responses at adjacent locations, which need a short-term memory and comparison circuit, which is plausible in the brain. This peak-seeking procedure requires listening to the sound source at least three times, which is slower than the midline-seeking procedure with bilateral LSO circuit.

Thirdly, we also extend learning beyond the azimuth plane, to learning a spherical localiser in both azimuth and elevation dimension. We use a compositional procedure, in which the spherical localiser is learned by combining the azimuth and elevation estimation procedures. The azimuth estimation can be done using any of the procedures above, while the elevation estimation can also be done using a similar procedure but in the elevation plane. Note that although the estimation is done separately for azimuth and elevation in the learning procedure, the final learned spherical localiser is represented by a single Student DNN. This procedure requires a estimation procedure implemented for each dimension. It also requires the agent to tilt its head in the elevation plane to estimate the elevation, in addition to rotating the head in the azimuth plane.

Finally, all the three above procedures are combined to train the Student model in a context where all the previous constrains are applied at the same time. Within the rotation range, the Student first uses the peak-seeking procedure in the azimuth plane to estimate the azimuth of the sound source, and then uses the peak-seeking procedure in the elevation plane to estimate the elevation of the sound source. The two estimations are combined to form the final parameter adjustment. Out of the rotation range, the Student uses the learned localiser to adjust the estimation, similar to the 2D case.

The results (Fig 7) show that the Student model can learn the spherical localiser with the overall mean absolute error of 3.1°, and a standard deviation 2.4°, with a restricted head rotation range and limited to a monaural Teacher circuit.

Here we present a particular procedure and monaural Teacher circuit, however it is clear from the construction that many possible learning procedures and circuits can be combined in diverse ways (e.g. by merging the maps learned from both left and right ears). Such flexibility is difficult to achieve within a supervised learning paradigm, which requires direct external supervision — such as the specific feedback from a fully developed visual spatial system. In contrast, bootstrapping enables this flexibility by removing the dependency on external supervision, thereby expanding the range of possible calibration signal sources and supporting context-dependent learning. As a result, it offers a useful first-step modeling tool for understanding learning in the brain, where multiple learning mechanisms can be selected or combined depending on the context.

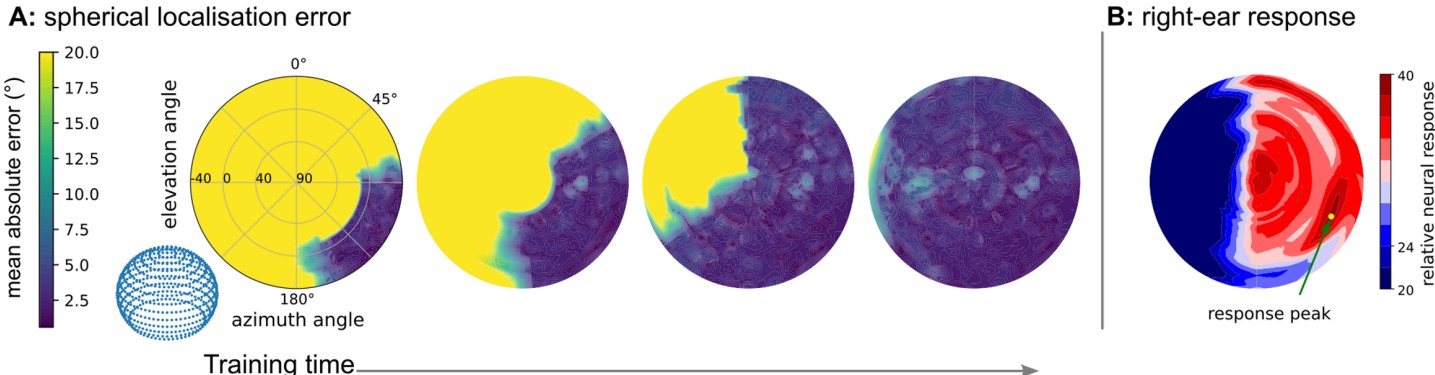

**Fig 7. Spherical localiser calibration with a limited head rotation range and a unilateral Teacher circuit.** (A) Evolution of the error in the learned localiser during training. Initially, only a small region around the peak response can be accurately estimated (blue region), but over training time the area of accuracy expands to cover the entire space. (B) The approximated 3D sound response of the right ear of the KEMAR manikin, showing the peak response at elevation=−10°, azimuth=115° on the spherical cap.

## Discussion

The acoustic cues underlying sound source localization can change during development, in response to events during a lifetime, and as a result of aging. Our auditory systems are able to adapt to these changes, but we do not yet know the learning mechanisms we use to do so. This learning problem is often assumed to be a passive process based on precise feedback from other systems, typically from the visual system. However, this is insufficient to explain learning during early development (for example a baby who has not yet learned to recognise visual objects and link them to their corresponding sounds), or for blind people who get no visual feedback at all. Here, we have shown that using an active process we refer to as *bootstrapping*, it is possible to learn to localize sound using only coarse grained feedback generated internally from the auditory system using robust innate mechanisms. Moreover, this approach allows for learning with much less supervision, even when this precise feedback is available. We showed that a range of specific models of learning are compatible with this framework, and indeed can be combined and composed in diverse ways. We suggest that different species, and even potentially the same individual in different contexts may rely on different strategies. We now briefly discuss some of the main hypotheses on learning spatial hearing, and then discuss the points raised in this paragraph in more detail.

### Robust learning via bootstrapping

Bootstrapping in this paper refers to a set of mechanisms by which the brain leverages coarse cues—such as detecting left-right differences, midline, or peak responses—to progressively learn a detailed sound localiser (Fig 8). Our main result is that bootstrapping using innate circuits is sufficient to serve as the calibration cue for learning spatial hearing. These innate circuits are simple, exist universally, and are effective across many contexts, making them a plausible foundational mechanism. Bootstrapping is also robust. Since innate and simple circuits (that we call Teachers) can provide the necessary coarse grained feedback in this framework, and these Teachers do not need to be perfect, the entire system also becomes

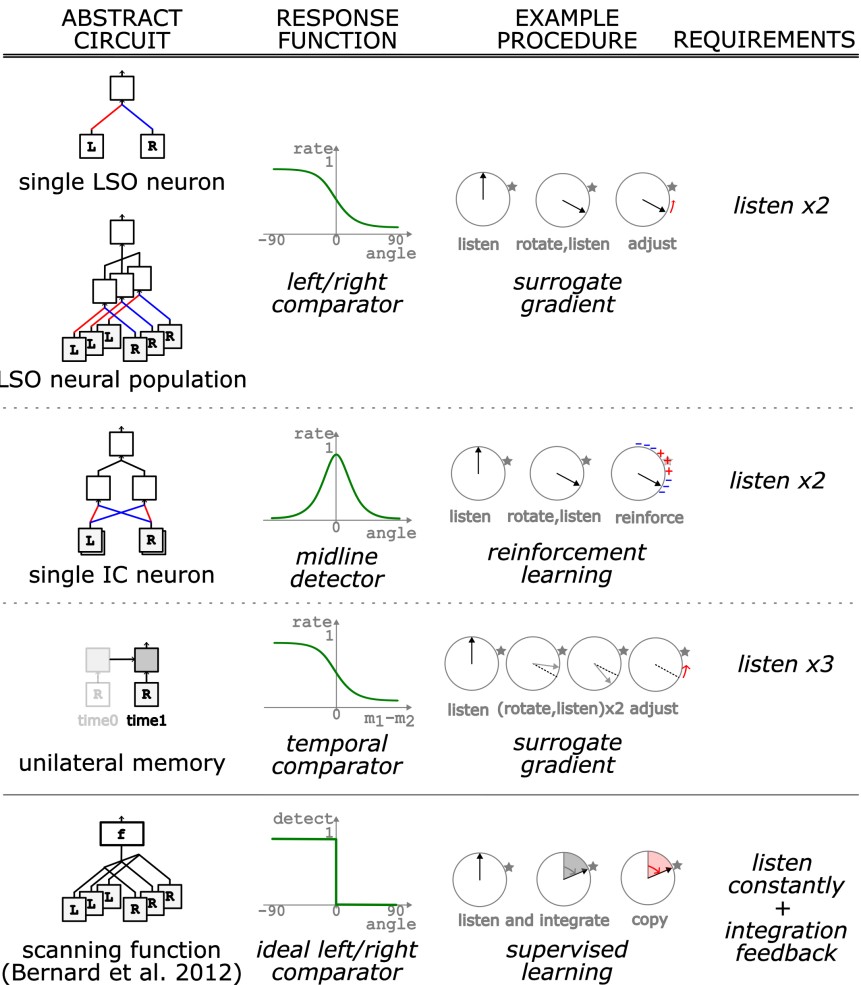

**Fig 8. Summary of the models.** Upper rows, models from this paper. Bottom row, model from [14]. Left column, abstract circuit: schematic representation of the circuit implementing the model. Next column, response function: tuning of the response of the Teacher circuit, showing different basic functions. Next column, example procedure: interactive learning procedures using the Teacher circuits. Right column, requirements: environment conditions need to be met to enable the Teacher and the procedure, here the main requirement is how many times the model needs to listen to the sound source.

resilient to changes in sensors, and adapts efficiently to different constraints, requiring minimal external supervision. The use of internally generated coarse grain feedback makes bootstrapping useful in the naturalistic condition where sensory cues are frequently changing and precise external feedback about errors is scarce.

## Multiple learning mechanisms

The brain likely employs a variety of teaching signals in different learning contexts, depending on availability, reliability or other criteria [23]. Even in normally-sighted individuals, visual-independent mechanisms likely operate alongside visual feedback [7] , compensating for vision's limited temporal resolution and spatial range [4].

Computational models like ours can help explore and predict the availability of a specific mechanism with given conditions. We have demonstrated that there is a rich collection

of learning methods that do not rely on precise external feedback. All of these methods are straightforward to implement in a biologically plausible neural circuit, although it is not necessary that the particular collections of mechanisms we have investigated are implemented in the brain as is. In addition, specific environmental conditions may be necessary to engage particular learning mechanisms. For example, adult barn owls show enhanced plasticity in their auditory space map only when hunting live prey but not receiving dead mice [37]. Similarly, mature ferrets only relearn sound localization after ear occlusion through active training in behaviorally relevant tasks - passive exposure to altered cues produces no adaptation even with normal animal-house sounds [12]. Furthermore, while ferrets can adapt to new auditory cues without external reward, initial behavioral training is still essential to enable such self-guided adaptation [12]. Although [12] proposes cue reweighting — where spectral cues compensate for occluded ITD/ILD cues — as one explanation, our bootstrapping model also offers an alternative: learning driven by intrinsically generated reward signals. A general algorithmic framework can thus bridge potential learning mechanisms with specific in vivo experimental contexts, making it possible to identify and isolate the contributions of individual mechanisms.

The existence of multiple learning mechanisms can also help explain and address individual differences in sensory adaptation. When faced with altered sensory conditions (e.g., prisms, hearing loss, cochlear implants), individuals show significant variation in learning speed and adaptation success [15,42]. Based on the experimental evidence and computational models discussed above, multiple factors influence adaptation outcomes, including learning environment, learning procedure design, and training frequency [12]. Understanding how different teaching signals emerge or are acquired may be crucial for explaining individual variation, as some mechanisms may not be innately available but require specific experiences to develop. While a recent qualitative framework represents a first attempt to predict learning outcomes based on individual contexts [42], computational models could further help predict adaptation trajectories and guide the design of personalized interventions. There are potentials to improve therapeutic procedures to improve spatial hearing after hearing loss, or accommodative training for hearing aid devices or cochlear implant. Current design strategies are often focused on understanding how to best provide direct and precise supervision, which our results suggest may not be well matched to how we learn, and therefore may not the most efficient approach. Although accurate individual learning result prediction remains a distant goal, computational models are advancing alongside other developments in related fields. Recent examples include wearable devices providing coarse tactile feedback about sound locations [43] and interactive training protocols encouraging active sound source manipulation [44]. These advances suggest opportunities for personalized training protocols based on individual needs and diverse learning mechanisms [45,46].

## Deep learning models

Data-driven machine learning models, particularly deep neural networks (DNNs), have become increasingly significant as computational modeling tools in neuroscience across various levels of analysis [17–19], including applications in the auditory pathway [20,21]. DNNs not only bear a resemblance to the architecture of biological neural networks but, perhaps more importantly, demonstrate the capacity to learn complex functions from data, analogous to the brain's ability to learn from experience. Their flexibility and computational power surpass traditional signal processing models [47–50], making them particularly well-suited for modeling brain functions at the behavioral level. In addition, DNN models and their learning algorithms are often task-agnostic, serving as a general modeling framework for diverse tasks

and data modalities. This flexibility enables the same modeling approach to generalize across a wide range of cognitive functions and neural systems, including vision, auditory processing, language, and motor control.

However, deep learning models do not respect known innate features of the biological systems they model. They typically have a homogeneous structure and are trained "end-to-end". This is good for high performance training but leads to one of the biggest shortcomings of DNNs as a neuroscientific model. In contrast to the brain, which seldom receives precise feedback (e.g. exact location information), training a DNN typically requires large amounts of meticulously curated data label [21,50]. This dependence becomes even more problematic when the data distribution shifts, as is common in animal life, requiring an entirely new data set to be gathered each time. This limitation is not only a mismatch with how the brain operates but also a significant practical hurdle in developing more adaptive and efficient intelligent systems. In machine learning terms, through evolution the brain has evolved an "inductive bias" that enables it to learn more with sparser data. We represent this in our model with a specific innate circuit structure that reduces the gap between a standard DNN and biology (e.g. we include the fact that newborn babies can orient towards sounds on the left and right), and guides complex function learning in an interactive environment without massive external supervision.

## Limitations and future work

We have shown that bootstrapping works in a range of conditions from fairly simple to quite challenging. However, our models do include some simplifications that would merit further study. For example, we do not use natural sounds but instead white noise throughout, and we do not include background noise, multiple sound sources, reverberation, etc. Our model of the auditory system is highly abstracted and simplified, and we only consider linear effects in our model of hearing loss. The actions that the model agents can take are very restricted. We only attempt to estimate direction of arrival and not distance. However, we do not expect that taking these into account would alter the core idea of the paper, as the simple innate Teacher circuits, such as the LSO-ensemble Teacher, are often robust to such variations. Future works could also incorporate additional acoustic cues, such as spectral features and interaural time differences, to model a more comprehensive and resilient spatial hearing system.

Our model is also not a perfect model of the brain. For a start, it uses an ungrounded multilayer perceptron network and has no spike timing information. We abstracted the Teacher and Student into separate independent modules, but in real brains it is unlikely that there is such a clear-cut distinction. Instead, our model should be considered at the behavioral level rather than at the implementation level. Questions about the biological plausibility of gradient descent or network stochasticity have also been left aside, as future progress in these areas can be incorporated into this framework. We also do not attempt to locate a specific pathway or region of the brain in which the proposed learning takes place. For example, we talked about how the LSO Teacher model could incorporate limited plasticity, but this plasticity does not necessarily have to take place in the LSO but in any number of related sites along this pathway, for example in the inferior colliculus which is documented to exhibit plasticity [51–53].

## Related models

There have been notable efforts to develop visual-independent models for sound source localization. For example, [13] proposed a sensorimotor learning model that maps high-dimensional acoustic features to low-dimensional spatial locations using an unsupervised manifold learning algorithm. In [14] the agent rotates its head toward the sound source and

uses motor feedback to supervise the sound localiser, by integrating the total rotation angle. Both models share similarities with our approach in that they are visual-independent and constructed a search procedure to generate the calibration signal. However, they rely on specific assumptions that differ from ours. The manifold learning algorithm [13] assumes a homogeneous relationship between changes of high-dimensional acoustic feature and changes of low-dimensional spatial location, and it requires uniformly sampled training positions to perform effectively. The model in [14] requires continuous acoustic input during the whole process of head rotation until the sound source angle is reached, while our bootstrapping model needs only two brief listening with small head movement. Such difference of algorithmic requirements can be critical for modeling animal behavior. In this specific case about head rotation, [12] measured the adaptation of ferrets to monaural occlusion with both long (1000 ms) and short (40 ms) sound stimuli. Head position tracking confirmed that 40 ms was insufficient for complete head rotations, while 1000 ms allowed full movements. The longer stimuli produced faster and more robust adaptation. While this learning process may involve other mechanisms beyond bootstrapping (e.g. extended evidence accumulation in neural pathways), it would be interesting to isolate their effects with the help of model simulations. Besides, in [14], the agent must integrate the total rotation angle as supervisory feedback for the auditory system. In contrast, our bootstrapping model requires the agent to stochastically execute the rotation order based on auditory predictions. Together, these models provide a more comprehensive perspective when distinguishing between the different interactions in an agent's auditory and motor systems, particularly with regard to the origin of calibration signals—whether they can be measured through auditory-to-motor mapping, motor efference copies, or afferent proprioception pathways [45]. In addition, taking a broader perspective, these models may also be viewed as special cases of bootstrapping, where each employs a distinct Teacher in a specific context.

## Methods

All code and trained network weights are available at the following GitHub repository: https://github.com/YangTrue/Learning-spatial-hearing-via-innate-mechanisms.

### Environment and agent

We simulated the interactive acoustic environment with an agent based on SLAB, a Python package for spatial audio [54]. The agent's acoustic features were modeled using standard KEMAR head-related transfer functions (HRTFs) that capture how the ears, head, and torso filter sounds from different spatial positions [55]. The default stimuli were 100 ms white noise bursts at 70 dB SPL presented in an anechoic virtual environment. Binaural signals received by the agent were generated by filtering the source sounds with the appropriate HRTF for each spatial location.

The spatial location of a sound source is described by vector $y$ in a head-centered coordinate system. With the origin at the center of the agent's head, $y$ connects the origin to the sound source position. The source distance $r = |y|$ is the magnitude of this vector. The elevation angle $\phi$ is measured between $y$ and the horizontal plane, while the azimuth angle $\theta$ is measured between the projection of $y$ onto the horizontal plane and the agent's forward-facing direction. Together, $y = (r, \theta, \phi)$ uniquely specify the sound source position relative to the agent. To reduce the complexity of the model and data required, and in line with many other studies in the field, sound source localization is simplified to determining the direction of arrival (DoA) of the sound, without estimating the source's distance from the listener.

Therefore the model is simplified to be $y = \theta$ in the 2D case in the azimuth plane, or $y = (\theta, \phi)$ for the 3D case.

The agent can rotate its head in azimuth ($\theta$) and elevation ($\phi$) around the fixed origin at the head center. The movement range varies by experimental context. Head rotation is guided by the agent's auditory spatial predictions.

The agent's spatial hearing was evaluated after learning by testing sound localization performance. In the same anechoic environment, white noise stimuli were presented from various positions. The agent reported the perceived source location using its learned localiser. Localization accuracy was quantified by the angular error between reported and actual positions.

**General model structure.** The general model structure is illustrated in Fig 9. A sound $S$ at position $y$ arrives at both ears. For the path to the left ear, the sound $S$ is passed through a head-related transfer function (HRTF), equivalent to a convolution of the sound with the corresponding head-related impulse response (HRIR):: $L^t = \text{HRIR}_{\text{left}}(y) * S$, where $\text{HRIR}_{\text{left}}(y)$ is the HRIR filter for angle $y$ at a fixed distance of 1.4 meters.

The KEMAR HRTF database [55] contains measurements at 710 pre-recorded source positions distributed across a spherical cap spanning elevations $\phi \in [-40°, 90°]$. For a queried source location $y$, the filter function is interpolated using the three nearest pre-recorded positions $\{\vec{p}_1, \vec{p}_2, \vec{p}_3\}$ that form a triangle containing $y$. The interpolated filter $H(y, v)$ at each frequency $v$ is computed using SLAB [54] as:

$$H(y, v) = \sum_{n=1}^{3} w_n H(\vec{p}_n, v)$$

where $w_n$ are the barycentric coordinates of $y$ with respect to $\{\vec{p}_1, \vec{p}_2, \vec{p}_3\}$, and $H(\vec{p}_i, v)$ are the measured filters at the triangle vertices.

Next, the cochleagram $L^{fb}$ is generated by applying an ERB filterbank: $L^{fb} = \text{ERB} * L^t$, to mimic cochlear frequency selectivity. The ERB filterbank consists of 24 channels per ear spanning 20 Hz to 20 kHz, with filter bandwidths following Moore & Glasberg's ERB formula [54,

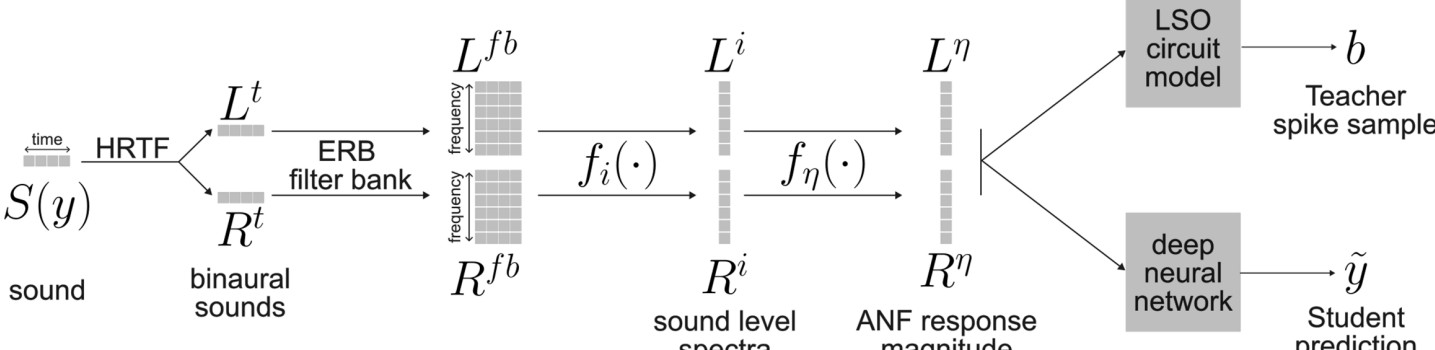

**Fig 9. General model structure.** A sound $S$ at location $y$ is first filtered via a pair of head related transfer functions (HRTF) to give the sounds $L^t$ and $R^t$ received by the two ears. Within each ear, a bandpass filter bank following the equivalent rectangular bandwidths (ERB) formula is applied to generate a cochleagram $L^{fb}$ and $R^{fb}$. The sound level for each frequency band is computed with the function $f_i$. In the general scenario, a sound level could be converted into a model of the auditory nerve fiber (ANF) response magnitude via a nonlinear function $f_\eta$, although in this paper we do not need a detailed model of the ANF response and therefore we simply use the identity function. Finally, the ANF responses are fed to the various different LSO and DNN circuits used.

56]. Each filter is cosine-shaped and centered at its peak frequency, with bandwidth scaling proportionally to center frequency.

The sound level for each frequency band $v$ is then calculated as: $L_v^i = f_i(L_v^{fb}) = 20\log_{10}(\text{RMS}(L_v^{fb})/2 \times 10^{-5})$, where RMS is the root mean square function over the time series.

Additional neural transformations of the peripheral auditory systems are simulated by $L^\eta = f_\eta(L^i)$, where $f_\eta(\cdot)$ simulates the auditory neural fiber (ANF), transforming the sound level spectrum to the relative response magnitude vector. In this paper, $f_\eta(\cdot)$ is chosen to be an identity function, simply preserving the relative magnitude of different sound intensities while its absolute scale is arbitrary.

The signal processing for the right ear follows a similar pathway. The resulting binaural magnitude vectors $(L^\eta, R^\eta)$ serve as input to both the Teacher circuit and Student network. While our model employs this simplified representation, more elaborate implementations could incorporate detailed neural response properties measured from the auditory system. The current level of abstraction is adequate for examining the principles of the spatial hearing process.

## Teacher neural circuit models

The Teacher circuit serves as the central focus of this study, which supports bootstrapping learning with basic innate mechanisms. While specific implementations vary by experiment, Teacher circuits follow the same principles: it is a simple neural circuit that can generates basic calibration signals during agent-environment interactions. Rather than directly providing precise calibration, the Teacher mediates the discovery of spatial information through structured exploration procedures. The same Teacher circuit may be combined with different procedures, depending on the specific requirements of the learning environment.

**Lateral superior olive data and model.** [31] recorded responses of neurons in the lateral superior olive (LSO) to different interaural level differences (ILDs) in cat's brain. Each neuron in this dataset is characterized by fitting two functions to its physiological recording [31]. The mean firing rate $\mu$ as a descriptive function of ILD $x$ follows:

$$\mu = a + \frac{b}{1 + \exp((c - x)/d)} \tag{3}$$

with response variability described by standard deviation: $\sigma = gx^h$ where $a, b, c, d, g, h$ are fitted parameters. This model enables simulation of LSO neural responses and their trial-to-trial variability for a given ILD value.

The input ILD $x$ for each left-side LSO neuron with characteristic frequency $v$ is calculated by $x_v = f_v(L^\eta) - f_v(R^\eta)$, where $f_v(\cdot)$ represents piecewise linear interpolation in the frequency domain. Similarly for $f_v(R^\eta) - f_v(L^\eta)$ is used for the right-side LSO neuron.

**Single LSO neuron circuit.** Single spike responses are sampled in two steps. First, a normalized firing rate $r$, the probability of spiking, is sampled based on the ILD response model (Eq 3) $\hat{r} \sim \mathcal{N}(\mu, \sigma^2)$, $r = \text{Norm}(\hat{r})$, with the normalization function $\text{Norm}(x) = \frac{x - x_{\min}}{x_{\max} - x_{\min}}$. Second, a binary spike or no-spike variable is sampled as $z \sim \text{Bernoulli}(r)$, modeling a stochastic neuron.

**LSO neural population circuit.** A population of 32 LSO neurons with different characteristic frequencies between 20 to 2200 Hz are recruited. A sigmoid readout neuron takes the normalized responses rate $r_i$ of each LSO neuron $i$ as input, and output a binary spike or no

spike variable $z_{\text{population}}$ by sampling. The overall computation of the circuit is:

$$\bar{r} = \frac{1}{32}\sum_{i=1}^{32} r_i, \tag{4}$$

$$\rho = \frac{1}{1 + \exp(-w\bar{r} - b)}, \tag{5}$$

$$z_{\text{population}} \sim \text{Bernoulli}(\rho). \tag{6}$$

The sigmoid function is initialized with $w = 10, b = -5$, approximating the identity function in the input range $[0, 1]$, in order to model a task-agnostic initialization. $w$ and $b$ can be adjusted to allow plasticity of the readout neuron.

**Midline detector.** We test a simple circuit model based on two LSO neuron population circuits. Let $z_L$ and $z_R$ be sigmoid outputs from left and right LSO neuron populations. The midline detector binary output variable $z_M$ is: $z_M \sim \text{Bernoulli}(r_M)$, where the spiking rate is $r_M = \text{Norm}(z_L \cdot z_R) \in [0, 1]$, the normalization function is $\text{Norm}(x) = \frac{x - x_{\min}}{x_{\max} - x_{\min}}$.

## Student neural network models

The Student is implemented with a deep neural network (DNN) as a behavioral level model for auditory space map or related neural systems, which we refer to as a "sound localiser". It takes binaural acoustic cues as input and predicts the sound source location as the output, which can be used to guide the agent's head movements . The network has 5 hidden layers of 128 units each with ReLU activations to allow for a complex non-linear mapping between input and output. Training uses stochastic gradient descent with the Adam optimizer [57], with default hyperparameters $\alpha = 0.001, \beta_1 = 0.9, \beta_2 = 0.999$, and batch size 32. Network parameters are initialized by sampling from a normal distribution. The implementation uses PyTorch [58]. This architecture remains the same across experiments, with only the Teacher's calibration signal varying between conditions, for better comparison.

## Learning algorithms

**Algorithm 1 Supervised learning with surrogate gradient method (Fig 2).**

```
 1: Initialize Student parameters θ
 2: Initialize learning rate α
 3: for episode i = 1 to N do
 4:     Sample sound xᵢ at random location yᵢ
 5:     Compute Student predicted location ŷᵢ = f(xᵢ, θ)
 6:     Rotate toward ŷᵢ and listen again
 7:     Compute Teacher predicted left/right feedback ỹ
 8:     Compute surrogate gradient ∇θLᵢ = ỹ ≈ sign(ŷᵢ − yᵢ)
 9:     Update Student parameters: θ ← θ − α · ∇θLᵢ
10: end for
```

**Bootstrapping with an LSO circuit (Fig 2).** For the response curves, we sampled each firing rate $r$ for 100 times at each spatial position ($[-90°, 90°]$ with $1°$ interval) for both the single LSO neuron and LSO neural population Teacher circuit, plotting mean responses (solid lines) with variance bars. The spike rate variance was quantified by drawing 100 binary samples $z$ from Bernoulli distributions for each sound presentation, with means determined by the sampled firing rates $r$. To measure the learning trajectories, we trained (Algorithm 1) a Student network to predict sound source directions between $-180°$ and $180°$. Training sounds

were presented with $70 \pm 20$ dB uniform random level variation to make the task more challenging, simulating sound source variations or distance changes. We evaluated performance every 2K steps during 160k total training steps (with learning rate linearly decay to 0 in the last 60K steps), measuring mean absolute error across uniformly spaced test positions at 5° intervals. This training-testing experiment was repeated 10 times with different Student network initializations.

**Measuring the effect of Teacher bias (Fig 3).** There are 32 Teacher circuits, each with a single LSO neuron from the [31] dataset with the characteristic frequency range from 20 to 2200 Hz. We simulated responses of each LSO using 10 sound presentations at each azimuthal location from –90° to +90° in 5° steps. For each neuron, the mean spike rate was calculated across the 10 presentations at each location to generate the average response curve. To determine the crossing point, we interpolated each mean response curve to identify where it intersected the midline (normalized firing rate equals 0.5).

Training was conducted over 100K steps and the learning rate linearly decays from $10^{-3}$ to 0 after 60K steps. We repeated each experiment 3 times with different random parameter initialization of the Student network.

**Cue disruptions (Fig 4).** Both Teacher circuit and Student networks received relative neural response magnitudes ($L = L^{\eta}, R = R^{\eta}$) of the auditory neural fibers as input (Fig 9). We simulated 3 different kinds of hearing disruptions by manipulating these input signals: shifting both L,R by a constant 20 ($L' = L - 20, R' = R - 20$)(equivalent to symmetrical hearing loss which reduces sound level by 20 dB) , scaling L,R by 0.5 ($L' = 0.5L, R' = 0.5R$) (simulating auditory neural fiber compression changes), or scaling L only ($L' = 0.5L, R' = R$) (unilateral disruption). To characterize network responses to these disruptions, we tested the Teacher across sound source angles from –90° to 90° in 1° steps, collecting 10 samples per angle to measure response variance. The Student was tested from –180° to 180° in 5° steps. For direct-recalibration experiments with each disrupted hearing condition, we retrained the Student using the Teacher's output for 100K steps, with initial learning rate $10^{-3}$ linearly decaying to 0 in the last 60K steps. We also performed a two-step calibration experiment where we first allowed plasticity in the Teacher circuit's readout neuron, training it as a binary left/right classifier by minimizing the binary cross-entropy loss (using only 2 randomly generated sound samples between –10° and 10°, gradient descent for 1000 steps with learning rate $3 \times 10^{-1}$), before using this recalibrated Teacher to train the Student again. All experiments are repeated 10 times with different random initializations of the Student network parameters. Both Teacher and Student were assessed again after recalibration using the same testing procedures described above.

**Reinforcement learning (Fig 5).** We characterized the Teacher circuit by recording 100 binary responses to sound samples at each 1° increment from –90° to 90°, calculating the mean response rate and variance. The Student network, representing the Gaussian policy ($\pi$) parameterized by $\theta$, was trained via bootstrapping using the policy gradient algorithm as described in Algorithm 2, where the Student network predicted the mean rotation direction ($\mu$) and maintained a fixed exploration variance ($\sigma = 3.6$). Training involved 200 K episodes with batch size 32, using a learning rate of $3 \times 10^{-4}$ that decayed linearly to 0 over the final 30K episodes. Each episode allowed maximum $T = 3$ roll-out steps, with rewards of +100.0 for successful localization (Teacher spike=1) and –5.0 penalties per step before success. During training, the agent stored transitions in episode memory to compute policy gradients, which were used to update network parameters by maximizing the expected cumulative reward function $J(\theta)$ with discounting factor $\gamma = 0.1$. This gradient based optimization process adjusted parameters to increase the probability of actions that led to higher rewards. We

**Algorithm 2 Policy gradient for sound source localization (Fig 5).**

```
 1: Initialize policy network (Student network) parameters θ
 2: Initialize learning rate α
 3: for episode = 1 to M do
 4:    Initialize episode memory E ← []
 5:    Sample sound stimulus at random location as initial state s₁
 6:    for roll-out step = 1 to maximum step T do
 7:       Sample predicted source location as action aₜ from Gaussian
   policy: aₜ ∼ 𝒩(μ(sₜ|θ),σ²)
 8:       Rotate toward aₜ. Observe reward rₜ according to Teacher
   circuit feedback. Update to next sound stimulus input state sₜ₊₁.
 9:       Store transition (sₜ,aₜ,rₜ,sₜ₊₁) in episode memory E
10:       End the episode if Teacher circuit spikes or the updated
   state sₜ₊₁ reaches out of range [−90°,90°]
11:    end for
12:    Compute policy gradients ∇_θJ(θ) ← Σ_{t=1}^T ∇_θ log π(aₜ|sₜ,θ) · Gₜ, where
   Gₜ ← Σ_{k=t}^T γ^{k−t}rₖ is the discounted return from time t
13:    Update policy network parameters: θ ← θ + α · ∇_θJ(θ)
14: end for
```

evaluated performance by testing the trained network at 5° intervals across the full range, repeating all experiments with 10 different network initializations.

**Unilateral teacher and spherical localiser (Figs 6 and 7).** The neural response of the right ear are measured with sound samples positioned at different locations. For the circular localiser (Fig 6F) on the azimuthal plane (elevation=0°), we sampled 360 positions from −180° to 180° in 1° increments. For the spherical localiser(Fig 7B), we sampled a grid of positions across −40° to 90° elevation and −180° to 180° azimuth in 1° increments, corresponding to the range of the KEMAR dataset. At each position, we measured the relative neural response $R^\eta$, averaged across 24 frequency channels to approximate overall monaural circuit activity. Maximum responses were observed at azimuth=−110° in the 2D case and at elevation=−10°, azimuth=115° in the 3D case. Importantly, the bootstrapping algorithm requires only a local maximum, within a local area around the right ear which has the same size as the rotation range, to guarantee learning by pointing the ear to the sound source. Neither the precise response magnitude nor maximum position affects final learning accuracy. Alternative methods using total energy of unilateral sound yielded similar response curves. While actual unilateral loudness perception in the brain would be more complex and individual-specific, current approximation is sufficient to work with the interactive bootstrapping procedure.

The maximum rotation range was restricted to $(−30°, 30°)$ in both azimuth and elevation dimensions, that is, the agent can only rotate from $y_0 = (0°, 0°)$ to $y' = (\phi, \theta)$ where both $|\phi| \leq 30°$ and $|\theta| \leq 30°$, simulating a small range of head rotation in the 3D space. The learned localiser represents positions relative to the peak response point, not the absolute position. For evaluation purposes, we computed the absolute angle by adding the relative angle to the right ear peak position (elevation=−10°, azimuth=115°). Bootstrapping training proceeded in two phases: First, we learned an initial localiser $M_1$ within the rotation range using the monaural circuit as Teacher, with training data sampled uniformly from relative range $(−30°, 30°)$ around the right ear for 300K steps. Second, we expanded the localiser beyond the fixed rotation range by multiple steps of self-bootstrapping, starting by replacing the innate Teacher by the learned localiser $M_1$. The training data and learned localiser ranges expanded progressively in both azimuth and elevation dimension in steps $i$, where each step results in a larger localisation range $M_i(\theta \in [\max(−30°i, −180°), \min(30°i, 180°)]$ in azimuth, $\phi \in [\max(−30°i, −40°), \min(30°i, 90°)]$ in elevation), until reaching $M_6$ which covers the full spherical cap within the azimuth $(−180°, 180°)$ and elevation $(−40°, 90°)$ range, with each

expansion step trained for 100K steps. To avoid under-fitting $M_1$ and propagating boundary errors in the following expansion steps, we extend the training steps for the boundary areas of $M_1$ with an additional 100 K refining steps using samples from range $(-40°, 40°)$ while limiting rotation to $(-20°, 20°)$. Weight duplication [59] is used during expansion, where we fixed parameters of the previous localiser while updating only the expanding localiser's parameters to prevent potential oscillation. The extended training scheme and the weight duplication do not require additional external labels but help to improve the learned accuracy and training speed (reducing the test mean absolute error by more than 5°). All experiments were repeated 10 times with different random initializations, using learning rate $10^{-3}$ with linear decay to 0 in the final 40% of training steps, and batch size 4. The Student network architecture remained the same with previous 2D experiments except for an additional output neuron for elevation angle in the final layer.

## Author contributions

**Conceptualization:** Yang Chu, Wayne Luk, Dan F. M. Goodman.

**Data curation:** Yang Chu, Dan F. M. Goodman.

**Formal analysis:** Yang Chu.

**Investigation:** Yang Chu, Dan F. M. Goodman.

**Methodology:** Yang Chu, Wayne Luk, Dan F. M. Goodman.

**Project administration:** Wayne Luk, Dan F. M. Goodman.

**Resources:** Dan F. M. Goodman.

**Software:** Yang Chu, Dan F. M. Goodman.

**Supervision:** Wayne Luk, Dan F. M. Goodman.

**Visualization:** Yang Chu, Dan F. M. Goodman.

**Writing – original draft:** Yang Chu, Wayne Luk, Dan F. M. Goodman.

**Writing – review & editing:** Yang Chu, Wayne Luk, Dan F. M. Goodman.

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
