## [Decision Letter · Decision Letter 0]

22 Jun 2025

PCOMPBIOL-D-25-00755

Learning spatial hearing via innate mechanisms

PLOS Computational Biology

Dear Dr. chu,

Thank you for submitting your manuscript to PLOS Computational Biology. After careful consideration, we feel that it has merit but does not fully meet PLOS Computational Biology's publication criteria as it currently stands. Therefore, we invite you to submit a revised version of the manuscript that addresses the points raised during the review process.

Please submit your revised manuscript within 60 days Aug 22 2025 11:59PM. If you will need more time than this to complete your revisions, please reply to this message or contact the journal office at ploscompbiol@plos.org. Please include the following items when submitting your revised manuscript:

We look forward to receiving your revised manuscript.

Kind regards,

Jens Hjortkjær, Ph.D.

Academic Editor

PLOS Computational Biology

Stacey Finley, Ph.D.

Section Editor

PLOS Computational Biology

**Additional Editor Comments:**

Your manuscript has been reviewed by three experts in the field. The reviews are generally positive but also highlight a need for different clarifications and include several suggestions for improvements. In your revision, please address each of the comments and suggestions given by the reviewers. One reviewer suggests adding additional references. You are free to follow this suggestion or not as you think best. Please note that review 3 is attached as a separate file.

**Journal Requirements:**

3) We noticed that you used the phrase 'data not shown' in the manuscript. We do not allow these references, as the PLOS data access policy requires that all data be either published with the manuscript or made available in a publicly accessible database. Please amend the supplementary material to include the referenced data or remove the references.

Potential Copyright Issues:

i) Figures 1, 2, 5, and 6. Please confirm whether you drew the images / clip-art within the figure panels by hand. If you did not draw the images, please provide (a) a link to the source of the images or icons and their license / terms of use; or (b) written permission from the copyright holder to publish the images or icons under our CC BY 4.0 license. Alternatively, you may replace the images with open source alternatives. See these open source resources you may use to replace images / clip-art:

6) Please provide a completed 'Competing Interests' statement, including any COIs declared by your co-authors. If you have no competing interests to declare, please state "The authors have declared that no competing interests exist". 

**Reviewers' comments:**

Reviewer's Responses to Questions

**Comments to the Authors:**

**Please note that one of the reviews is uploaded as an attachment.**

Reviewer #1: Summary

The manuscript proposes models for the calibration of auditory space without the use of visual cues (or alternative supervisory signals). Evidence is presented that this can be achieved using approximate feedback from a simple innate circuit able to distinguish left from right sound sources. Furthermore, supervised learning can be of benefit to maintain the adaptive neural representation. A number of possible mechanisms are discussed, which may interact in order to maintain an accurate auditory spatial map.

Main comments

The manuscript addresses the question of how humans calibrate auditory spatial hearing, in the absence of a supervisory signal such as vision. This is an important topic in the area of auditory spatial cognition, for which a number of alternative explanations have been proposed, but a satisfactory answer has not yet been established. The paper investigates “bootstrap learning” as a possible process for calibrating the auditory spatial map. Interesting and enjoyable to read, the ideas presented in the paper are intriguing, the methods appropriate, and the interpretation of the results is sound. The manuscript will be of interest to readers of PLOS Computational Biology. I have some comments that the authors may wish to consider, that mostly regard providing additional information and clarification. These are detailed below, which I hope will be of use to the authors.

Introduction: The Introduction is fairly short overall, and some additional information and background may help set the scene for the reader. Movement is a crucial element of the calibration process, as is made clear later on in the manuscript e.g. in Fig 2, (A) Interactive learning procedure with the left/right Teacher circuit. However, the text in the Introduction is not particularly clear regarding this. It is briefly mentioned on line 43 that “The “Agent”–defined here as any human, animal, or model capable of acting within the environment–receives auditory inputs and moves in the simulation,” but perhaps more emphasis could be placed on this aspect. The text appears to be discussing the role of audiomotor feedback to calibrate auditory space (e.g. as per Lewald, 2013, who defines audiomotor feedback as the “evaluation of systematic changes of auditory spatial cues resulting from head and body movements)”, but it is not entirely clear if this is the case, and it would help if the notion of audiomotor feedback was put into context with regards to the current work in the Introduction.

Lewald, J. (2013). Exceptional ability of blind humans to hear sound motion: implications for the emergence of auditory space. Neuropsychologia, 51(1), 181-186.

Regarding the question on line 9 (“What could be the calibration signal used by the brain to learn spatial hearing?”), some brief further information about this topic that has been discussed in the literature might strengthen the background for the reader. For example, calibration of auditory space might result from experience with how auditory spatial cues change with self-motion such as when approaching a sound or turning the head, or using audiomotor feedback from reaching out and touching sound producing objects (Kolarik et al., 2025), and it has been highlighted that information from alternative senses such as touch can help calibrate auditory space during development (Gori et al., 2014 – ref 36 in the manuscript).

Kolarik, A. J., & Moore, B. C. (2025). Principles Governing the Effects of Sensory Loss on Human Abilities: An Integrative Review. Neuroscience & Biobehavioral Reviews, 105986.

The manuscript assesses interesting variations of the learning procedures, including constraining the agent’s rotational range to a small window, a monaural Teacher circuit without relying on the bilateral acoustic cues, and extending learning to both the azimuth and elevation dimensions. Unless I have missed it, has a variation also been discussed that accounts for calibration of auditory space necessitated by changes in acoustic information as the head grows bigger from infancy to adulthood, and do changes in head size factor into the models discussed (e.g. resulting ILD changes)? The manuscript points out in the abstract that “The acoustic cues used by humans and other animals to localise sounds are subtle, and change during and after development,” and this issue has been discussed in the literature, therefore clarifying the answers to these questions in the text would be helpful.

Lastly, the manuscript discusses how spatial maps of sound source azimuth and elevation might be calibrated and maintained, however distance is not prioritised (line 596 “Typically, sound source localization is simplified to determining the direction of arrival (DoA) of the sound, without estimating the source’s distance from the listener”), and it is unclear why this is the case. Auditory distance is an important aspect of sound localization, and the issue of how this is calibrated (especially for blind individuals) is of current interest in the field. Perhaps some discussion of how the proposed models might help to calibrate auditory distance would make this aspect of the manuscript more complete.

Minor comments

Fig 1. Caption: The information here needs to be a little clearer, as the labelling and clarity of the text could be improved and will be difficult to follow for a reader unfamiliar with acoustics. For example, the caption reads: “The auditory space map need to be learned and continuously re-calibrated during lifetime. (A) Different learning paradigms and the overall model of the Agent for spatial hearing. The acoustic environment is simulated using pre-recorded head-related transfer functions, converted a spatial sound stimuli into a cochleagram…” For more clarity, small suggested changes could include: “The auditory space map needs to be learned and continuously re-calibrated over the individual’s lifetime. (A) Different learning paradigms and the overall model of the Agent for spatial hearing. The acoustic environment is simulated using pre-recorded head-related transfer functions (HRTFs), converting a spatial sound stimulus into a cochleagram…” The word cochleagram is mentioned in the caption and text, but unless I have missed it, it has not been defined. I would suggest explaining in a few words what this is to help the reader. Labels in the figure need explanations in the caption. I assume LF and HF refer to Low Frequency and High Frequency, but these need to be clarified. Similarly, does ILD refer to Interaural Time Difference here? This is mentioned later in the manuscript, but should be briefly mentioned in the caption when the reader is first introduced to it. DNN (deep neural network) should be specified in the caption so that the reader does not have to refer to the main text.

Fig 2: Some of the labels are very small and hard to read e.g. Small Teacher, Big student, or the small letters for L and R in panels B and C. I would suggest making them larger.

Fig3: Some of the light grey labelling makes it difficult to read (e.g. clockwise bias).

Fig 5: The labels SBC and GBC are very small.

Fig 6: The labels in panels f and h are very small.

Fig 8: Many of the grey labels within the plots (especially the numbers) are very small and hard to read.

Fig 9 caption: “Within each ear, a bandpass filter bank (ERB)…” Define ERB here.

Line 114: “gradient decent…” Typo? “gradient descent…”

ILD is defined twice on lines 120, and 130.

Reviewer #2: Spatial hearing has to be continually recalibrated due to differences in the geometry of the head and ears, particularly when these structures are growing, and when acoustic inputs are modified following hearing loss, etc. Because of its (normally) greater accuracy and reliability, many studies have focused on the role of vision in guiding auditory spatial processing, leading to the notion that a form of supervised learning takes place in the brain. While this is correct up to a point (principally because of evidence from studies of crossmodal plasticity in the superior colliculus during development), it is now well established that accurate sound localization can emerge without vision and that plasticity resulting from altered acoustic inputs is not dependent on (though may be enhanced by) concurrent visual information.

This study shows that a deep neural network can learn to localize sound without being provided with explicit teaching signals that might correspond to visual spatial cues and therefore stands apart from previous modeling studies in this area, which have been based on supervised learning. It also provides a potential framework for investigating the factors that give rise to the plasticity that has been demonstrated in psychophysical and neurophysiological studies. As such, I think this paper could be an important contribution.

The notion of “bootstrap learning” to establish accurate sound localization using an innate brain function – for example, neonatal orienting movements enabling left-right discrimination – is plausible. There are, however, various places where the assumptions and results deviate from the biology, raising questions over the feasibility of the strategies proposed. Also some of the terminology used in the paper is misleading.

Throughout the paper, the authors describe the DNN’s estimate of sound source location as an auditory space map. While it may be expedient to do so for describing the model outcomes, the only region of the brain where a coarse *map* of space is constructed is the superior colliculus, where it is superimposed on to a map of visual space. The auditory system itself (including auditory cortex and inferior colliculus, IC) does not represent sound source location topographically, so I would recommend that the authors consider using alternative terminology (e.g. the network's representation or estimate of sound source location) to avoid misleading readers.

Why was the lateral superior olive selected as a Teacher circuit? Interaural time differences are often claimed to be the dominant acoustic cue used by humans to localize sound, so some justification for why the LSO was selected rather than the medial superior olive would be helpful.

The statement in second paragraph on page 2 (lines 27-38) about the accuracy of “innate neural circuits” needs more explanation (and references).

The term “spatial sound” is used in several places. The word “spatial” doesn’t add or mean anything informative (“spatialized” or “externalized” might be better but none of them are necessary in the contexts used).

The absolute errors produced by the student network are very small and uniform across azimuth. This is very different from real localization accuracy, which varies greatly with sound source location. This shortcoming should be discussed.

Line 156: “single-LSO circuit”. Line 161: “single LSO”. What does that mean? One side of the brain?

A strength of the model is the capacity to manipulate the inputs to try to replicate different types of hearing loss. Although the effects of the manipulations shown in Figure 4 are interesting, I am not convinced that bilateral/unilateral scaling of ILD sensitivity accurately reflects loss of non-linear compression in the ear and therefore whether this is a useful transformation to incorporate in the model. The first paragraph on page 10 has no references to sport this. Unilateral shifting would have provided a closer match to the many experimental studies of unilateral conductive hearing loss.

Incorporating limited plasticity in the LSO Teacher enabled recalibration of the student network following these manipulations but is there any evidence that this is biologically plausible? How might this be manifest in the LSO?

The model architecture is well designed. This included a stage in which the broadband sound was filtered by left and right ear HRTFs. How important is this step, particularly as the authors say that LSO neurons are not sensitive to these spectral cues? Could the spectral cues (a single HRTF) provide a teacher signal for calibrating binaural/azimuthal sensitivity (in the IC or at higher levels of the auditory pathway)? This possibility has been proposed in studies in which spectral cues have been manipulated.

The authors suggest that one source of “intrinsic reward” signal might be neurons tuned to midline sound sources (zero ILD), stating that this is consistent with “observed spatial tuning curves in the IC” [32-35] (lines 333-334). Only one of these 4 cited papers is about the IC and even then the statement is incorrect. The great majority of IC neurons have contralateral preferences and almost none are tuned to the midline. The modeling therefore predicts a feature that is shown by very few neurons, casting doubt over its feasibility.

The terms 2D and 3D space are used too loosely. The directional coordinates of a sound source are defined by the two dimensions of azimuth and elevation, whereas distance provides the third dimension. Very reasonably, distance is fixed to simply the modeling but using 2D and 3D space to refer to azimuth and azimuth plus elevation, respectively, is misleading.

There are a few typos in the manuscript. These include but are limited to:

Lin 199: “due the”

Line 598: “model is simplify to be”

Line 781: “sphereical”

Reviewer #3: Review is uploaded as an attachment

**Have the authors made all data and (if applicable) computational code underlying the findings in their manuscript fully available?**

Reviewer #1: Yes

Reviewer #2: Yes

Reviewer #3: Yes

PLOS authors have the option to publish the peer review history of their article (what does this mean?). If published, this will include your full peer review and any attached files.

Reviewer #1: No

Reviewer #2: No

Reviewer #3: No

**Figure resubmission:**
---

## [Decision Letter · Decision Letter 1]

21 Sep 2025

Dear chu,

We are pleased to inform you that your manuscript 'Learning spatial hearing via innate mechanisms' has been provisionally accepted for publication in PLOS Computational Biology.

Best regards,

Jens Hjortkjær, Ph.D.

Academic Editor

PLOS Computational Biology

Stacey Finley, Ph.D.

Section Editor

PLOS Computational Biology

Reviewer's Responses to Questions

**Comments to the Authors:**

Reviewer #1: I appreciate the efforts of the authors in addressing the comments from the previous round of reviews. The updated version of the manuscript is sound, and I have no further comments.

Reviewer #2: The authors have addressed my previous comments. The expression 'learn a sound localiser', which is now used in a few places, is a little odd. It's up to you but you might consider 'learn to localise sound' instead.

Reviewer #3: Thanks for your answers and adjustments to the manuscript. I've read your responses to me and the other reviewers. And I'm happy to recommend acceptance.

I have one minor comment regarding Q5 in my original review (Midline detector as intrinsic reward). I'd hoped for an explanation for "how" the binary reward scheme provides sufficient learning signal. Maybe the authors are arguing here that across a batch of examples you get some signal perhaps? What happens on a single-trial level?

**Have the authors made all data and (if applicable) computational code underlying the findings in their manuscript fully available?**

Reviewer #1: Yes

Reviewer #2: Yes

Reviewer #3: Yes

PLOS authors have the option to publish the peer review history of their article (what does this mean?). If published, this will include your full peer review and any attached files.

Reviewer #1: **Yes: **Andrew Kolarik

Reviewer #2: No

Reviewer #3: No

---

## [Editor Report · Acceptance letter]

PCOMPBIOL-D-25-00755R1

Learning spatial hearing via innate mechanisms

Dear Dr chu,

I am pleased to inform you that your manuscript has been formally accepted for publication in PLOS Computational Biology. Your manuscript is now with our production department and you will be notified of the publication date in due course.

With kind regards,

Olena Szabo
